# Screening the Carbon Source Type in Solid-State Fermentation with *Phanerochaete chrysosporium* to Improve the Forage Value of Corn Straw and Rice Straw

**DOI:** 10.3390/ani13050888

**Published:** 2023-02-28

**Authors:** Ying Wang, Jia Yu, Qiang Li, Junfang Zhang, Sajida Naseem, Bin Sun, Lin Tang, Seongho Choi, Xiangzi Li

**Affiliations:** 1Engineering Research Center of North-East Cold Region Beef Cattle Science & Technology Innovation, Ministry of Education, Department of Animal Science, Yanbian University, Yanji 133002, China; 2Department of Animal Science, Chungbuk National University, Cheongju 28644, Republic of Korea

**Keywords:** solid-state fermentation, *Phanerochaete chrysosporium*, in vitro straw fermentation

## Abstract

**Simple Summary:**

The addition of carbon sources to white rot fermentation straw can promote the decomposition of organic matter by white rot fungi, shorten the fermentation cycle, and help retain more nutrients in the straw feed. Carbon sources (glucose, sucrose, molasses, or soluble starch) were added in 21 days of fermentation of corn and rice straw by white rot fungus (*Phanerocacee pusosporium*). The results showed that the molasses and glucose as carbon sources showed the greatest enhancement in the nutritional value of maize and rice straw after 14 days of solid-state fermentation (SSF) treatment.

**Abstract:**

Poor quality straw can be made more digestible and palatable through delignification using white rot fungi as a biological treatment in SSF. The decomposition of organic matter by white rot fungi is improved when a carbon source is added. Reducing the fermentation cycle can also help retain more nutrients in straw feed. To increase rumen digestibility and nutrient utilization, corn straw and rice straw were subjected to SSF with white rot fungi (*Phanerochaete chrysosporium*) for 21 days. The type of carbon source (glucose, sucrose, molasses, or soluble starch) was optimized, and the nutrient composition and in vitro fermentation parameters of the fermented straw were assessed. In the fermented corn straw and rice straw supplemented with different carbon sources, the results showed a decrease in lignin content, dry matter, cellulose, and hemicellulose loss, and an increase in crude protein content after 21 days. Total volatile fatty acid and ammonium nitrogen concentrations increased significantly (*p* < 0.01) during in vitro fermentation. Overall, the most enhanced nutritional values for corn straw and rice straw were observed after 14 days of SSF in the groups using molasses or glucose as a carbon source.

## 1. Introduction

Crop waste, including straw, hulls, and husks, is often stacked or burned causing environmental pollution. China is a sizable agricultural nation with a focus on grain production, and the enormous amount of straw generated leads to severe environmental pollution. Statistics indicate that 7.35 × 10^11^ kg of straw was fed to agriculture animals in China in 2021, demonstrating the enormous development potential of straw feed [1]. Crop straws are the primary source of roughage for ruminants; however, their fiber digestibility in the rumen typically does not exceed 50% [2]. Approximately 40–50% of the fiber in legumes and 60–70% of the fiber in grass can be digested by ruminants; however, some fibres in the straw, such as lignin that cannot be digested, regardless of how long they remain in the rumen. By functioning as a physical barrier and cross-linking with polysaccharides via ferulic acid bridges, lignin is hypothesized to prevent the microbial degradation of fibrous polysaccharides [3]. Various pretreatment techniques have been employed to overcome lignin barriers, including physical (grinding and irradiation), chemical (acids, bases, oxidation, and organic solvents), physicochemical (extrusion, hydrothermal, and steam blasting), and biological (microbial and enzymatic) techniques [4,5,6,7]. However, these technologies have several drawbacks, such as high cost, high energy need, economic unattainability, and environmental unfriendliness [6]. Biological pretreatment of ruminant roughage with fungi offers a low-energy and secure substitute to avoid these drawbacks [8,9,10].

White rot fungi offer the greatest potential among naturally existing fungi for selective lignin removal. Since Kirk and Moore (1972) showed that white rot fungi may selectively degrade lignin and improve rumen degradation of swamp and birch trees, numerous researchers have worked to increase the digestibility and nutritional value of lignocellulosic resources using white rot fungi. Poor quality straw can be made more digestible and palatable through a biological delignification process using white rot fungi in solid-state fermentation (SSF) [11,12]. The degradation of neutral detergent fibers (NDF) in ruminal straw was found to increase by 13% in vitro when fibrolytic enzymes from several white rot fungi were added to wheat straw [13]. To increase in vitro gas production from wheat straw, Nayan et al. (2018) treated various wheat straws with the white rot fungi *Ceriporiopsis subvermispora* and *Lentinula edodes* [14]. Shrivastava et al. (2011) improved wheat straw digestibility and nutrient content through SSF for 15 days with *Ganoderma* sp. *rckk* [15].

White rot fungi may be used to increase the digestibility and nutrient utilization of roughage in the rumen. The degradation mechanisms have not yet been fully elucidated because the structures are complicated and numerous enzymes, free radicals, and tiny molecules are involved in the degradation of lignin by white rot fungi [16]. Although there are many different types of white rot fungi, most grow slowly, are marginally resistant to weeds, degrade lignin, and produce large amounts of hemicellulose and cellulose. These characteristics result in a lengthy process for white rot fungi to ferment straw and necessitate sterilization conditions, both of which severely restrict the promotion and use of white rot fungi for straw fermentation [17]. Additionally, while white rot fungi specifically break down lignin during the nutritional growth phase, they can also extensively degrade hemicellulose and cellulose [18]. As a result, the incubation period has a significant impact on the selectivity of white rot fungi. Liang et al. (2010) demonstrated that the addition of 0.007% rhamnolipid disaccharide to *Phanerochaete chrysosporium*-fermented straw increased lignin peroxidase activity by 86% and lignin degradation by 54% [19,20].

In summary, in order to investigate whether the addition of different carbon sources and reduced fermentation cycles could improve the ability of white rot fungus to degrade organic matter and reduce nutrient losses. A common strain of white rot fungus, *P. chrysosporium* [21], and corn straw and rice straw were chosen in this experiment for a 21-day solid-state fermentation with different carbon sources (glucose, sucrose, molasses, or soluble starch) in order to increase the quality of straw fermentation by white rot fungi. The aim of this study was to determine the optimal fermentation conditions for corn and rice straw with white rot fungi to improve nutrient availability and digestibility for use in ruminant feeds.

## 2. Materials and Methods

### 2.1. Treatment of Corn Straw and Rice Straw

Corn straw and rice straw were produced in the town of Chaoyangchuan, Yanji, China. The straw was dried naturally, cut into 3–5 cm pieces, crushed in an ultrafine grinder, and passed through a 60-mesh sieve (0.25 mm aperture) for further use. The straw was then treated with *P. chrysosporium* (CICC 14076, China Centre of Industrial Culture Collection, Beijing, China) with a concentration of 2 × 10^8^ cfu/mL. As this experiment was carried out using dried maize straw, the substrate itself had a low water content, so a fermentation broth of 70% of the mass of the fermented substrate was added. For the control group, 25 g of corn straw or rice straw was added to 15 mL of the fermentation solution (volume of 10% white rot fungal mixture and 0.1% urea in distilled water). Solutions including different carbon source types (A: glucose; B: molasses; C: sucrose; and D: soluble starch) (Laboratory conservation) were mixed with 25 g of corn straw or rice straw to make up the eight treatment groups. Each solution contained 3% of the respective carbon source, 0.1% urea, and 10% white rot fungus in 15 mL of distilled water.

### 2.2. In Vitro Fermentation Experimental Design

A one-way experimental design was used to investigate the effects of different carbon sources and white rot fungi on the rumen fermentability of SSF straw over 21 days. After mixing the corn straw and rice straw with their respective fermentation solutions, they were placed in a plastic bag and kept at room temperature (28 °C) for 21 days for fermentation. Samples were collected every 7 days. Corn straw and rice straw were divided into control group and four treatment groups (A: glucose; B: molasses; C: sucrose; D: soluble starch) and samples were collected at 0, 7, 14, and 21 days. Every group with three parallel groups each were subjected to in vitro fermentation using 1 g of straw as substrate, and the reaction was terminated after 72 h. Determination of fermentation indicators pH, NH_3_-N, IVDMD and VFA. For the in vitro digestion test, 1 g of fermentation substrate was accurately weighed, poured into a fiber filter bag, sealed, and packed into a pre-warmed (39 °C) in vitro digestibility tube. An automatic dispenser was used to add 70 mL of mixed artificial rumen culture solution [22] to the digestibility tube, which was then mixed and immediately incubated in a thermostatic water bath incubator (39 °C) for 72 h. The fermentation mixture was then centrifuged at 12,000× *g* for 10 min at 4 °C. The resulting supernatant was used to determine pH and volatile fatty acid (VFA) [23] and ammonia nitrogen (NH_3_-N) [24] concentrations. The pH was determined using a pH meter, and NH_3_-N was determined using the phenol-sodium hypochlorite colorimetric method [24]. The organic acids were determined using a TC-C18 column (250 mm × 4.6 mm × 5 µm; Agilent Technologies, Santa Clara, CA, USA) at 50 °C with methanol as mobile phase A and 0.01 mol/L aqueous KH_2_PO_4_ (pH adjusted to 2.70 with phosphoric acid) as mobile phase B at a flow rate of 0.7 mL/min. A total of 5 mL of straw filtrate was centrifuged and mixed with 1 mL of 25% metaphosphoric acid; 1 mL of this mixture was centrifuged at 4 °C for 15 min at 12,000× *g* and then the supernatant was filtered through a 0.22 µm aqueous membrane. The organic acid concentration was immediately determined using a 25 µL automatic injection needle (Agilent Technologies, Santa Clara, CA, USA) with a volume of 5 µL, calculated using an external standard. In addition, the 72 h fermentation substrate was dried at 105 °C for 8 h to assess the dry matter digestibility [25]. In vitro digestibility of dry matter (NH_3_-N) was calculated using the following equation: IVDMD (%) = (sample dry matter weight − residue dry matter weight)/sample dry matter weight × 100.

### 2.3. Chemical Analysis

The dry matter (DM) content of the samples was determined via oven drying at 105 °C for 16 h (method G-003/1). The N concentration of milled straw was determined using the Kjeldahl method, and crude protein (CP) was calculated as N × 6.25. Ash content was measured after treatment of the samples in a muffle furnace at 600 °C for 6 h [26]. Ash-free neutral detergent fiber (ANF) [27], ash-free acid detergent fiber (ADF), and lignin (Van Soest 1973) were determined as previously described. ADL is the residue of ADF minus ADF digested by 72% sulfuric acid and then subjected to ashing. Hemicellulose content was calculated as the difference between NDF and ADF, and cellulose was calculated as the difference between ADF and acid detergent lignin (ADL).

### 2.4. Statistical Analysis

The trial data were collated in Microsoft Excel (Microsoft, Redmond, WA, USA) and analyzed by one-way analysis of variance (ANOVA) using SPSS 21.0 (IBM, Armonk, NY, USA). Differences were considered statistically significant if *p ≤* 0.05.

## 3. Results and Discussion

DM, ADF, and NDF concentrations in straw typically decrease, while CP content rises, when straw is fermented by white rot fungi [28,29]. According to this study’s findings, addition of carbon sources for corn and rice straw fermentation decreased dry matter, ADF, and NDF losses, while increasing protein content when compared to the respective control group. When straw was supplemented with any of the four different carbon sources, the CP content of the straw increased over time and was significantly higher than it was at 0 days (*p* < 0.01), but the DM content of all groups declined as fermentation time increased (*p* < 0.01) (Table 1). These results were similar to those reported by Zhao et al. (2020) [29]. According to previous reports [29], one cause of DM loss is the secretion of enzymes by white rot fungi during asexual growth, which break down macromolecules in corn stover to release carbon and nitrogen. The increase in protein content is likely due to the conversion of the carbohydrates degraded by the white rot fungi into CO_2_ and H_2_O. Hence, the increase in CP content and decrease of NDF and ADF content in the substrate with increasing fermentation time. The NDF and ADF content in the substrates significantly decreased in this study (*p* < 0.01; Table 1 and Table 2), which is consistent with a previous report by Wang et al. (2021) [30]; supplementation with a carbon source reduced this loss. The order of ability of the different carbon source groups to increase the CP content of corn stover was as follows: molasses > glucose > sucrose > soluble starch > control. The ash was composed of mineral bars and inorganic matter. While there was no significant change in the ash content between the control and glucose groups after 14 days of fermentation (*p* > 0.05), the ash content of the soluble starch, molasses, and sucrose groups increased significantly with increasing fermentation time (*p* < 0.01), and was significantly higher in the molasses group than in the other groups at 14 days (*p* < 0.01; Table 1). The order of ability of the different carbon source groups to increase the CP content of the rice straw was as follows: glucose > molasses > control > sucrose > soluble starch. The ash contents of rice straw in the added carbon source groups were significantly higher than that of the control group after both 14 and 21 days of fermentation (*p* < 0.01; Table 2).

Changes in the chemical composition of straw caused by fungi have been reported [31,32]. The results of this study showed an increasing loss of Cellulose, Hemicellulose, and Lignin in all corn straw and rice straw treatment groups during the 21 days of fermentation (Table 3 and Table 4), and the difference was significant (*p* < 0.01), but the loss in control group was greater than that in the four treatment groups (A: glucose; B: molasses; C: sucrose; D: soluble starch). White rot fungi can produce a wide range of lignocelluloses. The complete degradation of cellulose, hemicellulose, and lignin during fermentation results in a significant loss of carbohydrates [33]. In this study, the loss rates of cellulose, hemicellulose, and lignin were calculated to assess which carbon source could reduce the loss of hemicellulose and cellulose from corn stover and enhance the degradation of lignin by the white rot fungus (Figure 1 and Figure 2). At the 14th day of fermentation, the loss rate of cellulose and hemicellulose of corn straw were significantly higher in control group than the other treatment groups (A: glucose; B: molasses; C: sucrose; D: soluble starch; Figure 1A,B). The lignin degradation rate in the molasses group was higher than that in the sucrose, glucose, soluble starch, and control groups (Figure 1C). The fiber loss in rice straw was greatest in the soluble starch group followed by the control, sucrose, molasses, and glucose groups. Hemicellulose loss was greatest in the soluble starch group followed by the control, molasses, sucrose, and glucose groups. Lignin degradation was highest in the sucrose group than in the glucose, molasses, and control groups, and lowest in the soluble starch group (Figure 2C). The molasses and glucose groups showed the lowest loss of hemicellulose and cellulose and the highest degradation of lignin in corn straw and rice straw fermented by white rot fungi. It has been shown that carbon sources can control the secretion of polysaccharide-hydrolytic enzymes by wood-rotting fungi [33] and that ligninase production by white rot fungi can be effectively stimulated by glucose supplementation, with continuous low concentrations of glucose supplementation achieving better enzyme production [34]. In this study, the lignin degradation rates of the carbon supplementation groups were higher than those of the control group, and the loss of cellulose and hemicellulose was reduced. Therefore, it is reasonable to speculate that the addition of a carbon source to straw fermentation with white rot fungus could enhance the production of plasmalogenase and thus enhance the selective degradation of lignin.

IVDMD is an important index of nutrient value of roughage, and it is mainly affected by cellulose content and lignification degree. In general, the lower the degree of lignification, the higher the IVDMD value and the better the quality of roughage [35]. The results of this experiment showed that the IVDMD of the added carbon source groups were higher than that of the control group at the same fermentation time (*p* < 0.01; Table 5 and Table 6). Additionally, the IVDMD of all groups increased with time and was significantly higher on day 21 than on day 0 (*p* < 0.01; Table 5 and Table 6). This was consistent with the observed changes in lignin content. At the 14th day of fermentation, the IVDMD value of corn stover in the molasses-added group was twice that of control group (*p* < 0.01; Table 5). These results were consistent with those reported by Datsomor et al. (2022) [34]. In the rumen, VFAs are produced through the fermentation of dietary carbohydrates; they provide 70–80% of the energy for ruminants, reflect the nutritional value of the feed and the state of rumen fermentation, and are mainly influenced by feed NDF content [35,36]. In the present study, VFA concentrations peaked at 14 days of fermentation in all corn straw and rice straw treatment groups (*p* < 0.01; Table 7 and Table 8), and were significantly higher in the groups with added carbon sources than in the control group at the same fermentation time. The glucose group had a significantly higher VFA concentration than all other groups at 14 days of fermentation (*p* < 0.01; Table 7 and Table 8). Acetic acid is a major source of energy for ruminants [37] and propionic acid must be present to make full use of acetate in fat synthesis. Therefore, both these acids affect the digestion and metabolism of ruminant diets. In all treatment groups, acetic acid content of corn was highest at 14 days of fermentation and then decreased (*p* < 0.01; Table 7 and Table 8), while propionic and butyric acid content gradually increased over the 21 days of fermentation (*p* < 0.01; Table 7 and Table 8). The molasses group had higher proportions of acetic, propionic, butyric, and valeric acids than all other treatment groups after 14 days of fermentation (*p* < 0.01; Table 7 and Table 8). pH and NH_3_-N are important indicators of the rumen environment. They are used to regulate the homeostasis of rumen environment, and NH_3_-N can be absorbed and utilized by rumen epithelial cells. In this study, all groups with carbon source additives had a higher pH at day 14 of fermentation than the control group, and the values observed were within the normal range for optimal rumen metabolism (5.5–7.5). In all treatment groups, NH_3_-N concentrations peaked at 14 days of fermentation (*p* < 0.01; Table 7 and Table 8) and then gradually decreased corn, with higher NH_3_-N concentrations observed in the groups with carbon sources than in the control group. This study showed that rumen microbial activity is positively correlated with the CP content of the rumen fermentation substrate. An increase in microbial activity has been shown to increase the NH_3_-N concentration and promote the synthesis of rumen microbial proteins [38], which is consistent with the results of the present study. There was a decrease in NH_3_-N concentrations at 14–21 days in this experiment, which is unlike the previous report by Thao et al. (2014). Therefore, it is reasonable to speculate that the addition of a carbon source to straw fermentation by the white rot fungus could improve the rumen digestibility and nutritional value of the straw. This experiment mainly studied the influence of this treatment method on the composition and in vitro digestibility of straw, but did not explore the changes of metabolites and differences of straw degradation products after white rot fungus added carbon sources, and In-depth exploration of the fermentation mechanisms of white rot fungus.

## 4. Conclusions

Addition of carbon sources to SSF with *P. chrysosporium* reduced the loss of DM, ADF, NDF, cellulose, and hemicellulose and increased the CP content of corn straw and rice straw. Maize and rice straw in the molasses and glucose groups exhibited significantly improved digestibility and forage quality in vitro, with the highest forage value at the 14th day of fermentation. This study determined the optimal fermentation time and carbon source type with white rot fungi in corn and rice straw to improve the utilization and digestibility of nutrients for ruminant feed.

## Figures and Tables

**Figure 1 animals-13-00888-f001:**
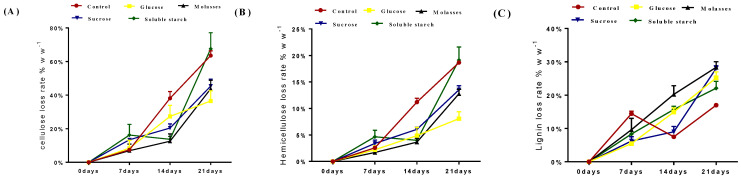
Loss rate of cellulose, hemicellulose, and lignin from corn stover during 21 days of solid–state fermentation with *Phanerochaete chrysosporium*. (**A**) Cellulose loss rate. (**B**) Hemicellulose loss rate. (**C**) Lignin loss rate.

**Figure 2 animals-13-00888-f002:**
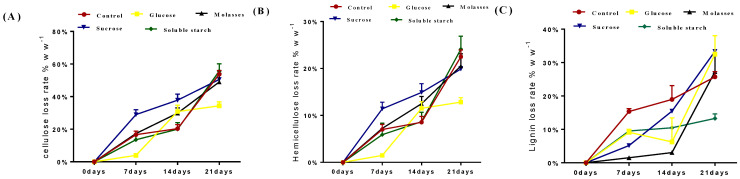
Loss rate of cellulose, hemicellulose, and lignin from rice straw during 21 days of solid–state fermentation with *Phanerochaete chrysosporium*. (**A**) Cellulose loss rate. (**B**) Hemicellulose loss rate. (**C**) Lignin loss rate.

**Table 1 animals-13-00888-t001:** Chemical composition of corn straw fermented by white rot fungi with the addition of different carbon sources (g/kg DM) after 0, 7, 14, and 21 days.

Item	Treatment	Sampling Day				SEM	*p*
		0	7	14	21		
DM	Control	80.56 ^a^	74.18 ^b^	63.43 ^c^	59.88 ^c^	2.534	<0.01
	Glucose	84.95 ^a^	73.16 ^b^	62.32 ^c^	59.76 ^c^	3.210	<0.01
	Molasses	86.39 ^a^	74.76 ^b^	64.31 ^c^	63.58 ^c^	2.909	<0.01
	Sucrose	85.49 ^a^	72.36 ^b^	62.2 ^c^	60.94 ^c^	3.010	<0.01
	Soluble starch	85.09 ^a^	71.59 ^b^	62.58 ^c^	62.58 ^c^	2.804	<0.01
	SEM	0.725	0.621	0.985	0.534	_	_
	*p*	NS	NS	NS	NS	_	_
CP	Control	2.41 ^d^	2.88 ^c^	3.17 ^Cb^	3.31 ^Ba^	0.10	<0.01
	Glucose	2.31 ^d^	3.05 ^c^	3.54 ^Ab^	3.90 ^Aa^	0.18	<0.01
	Molasses	2.18 ^d^	3.00 ^c^	3.48 ^Ab^	3.90 ^Aa^	0.19	<0.01
	Sucrose	2.23 ^c^	3.08 ^b^	3.21 ^BCab^	3.41 ^Ba^	0.140	<0.01
	Soluble starch	2.40 ^c^	3.03 ^b^	3.33 ^Ba^	3.41 ^Ba^	0.122	<0.01
	SEM	0.04	0.03	0.04	0.07	_	_
	*p*	NS	NS	<0.01	<0.01	_	_
Ash	Control	5.78 ^A^	5.54	5.94 ^AB^	5.49	0.088	NS
	Glucose	4.69 ^BC^	5.41	5.54 ^AB^	5.63	0.134	NS
	Molasses	4.62 ^BCb^	5.52 ^a^	5.78 ^ABa^	5.84 ^a^	0.153	<0.01
	Sucrose	5.13 ^Bb^	5.51 ^ab^	6.12 ^Aa^	5.51 ^ab^	0.128	<0.05
	Soluble starch	4.40 ^Cc^	5.63 ^a^	5.07 ^Bb^	6.01 ^a^	0.187	<0.01
	SEM	0.137	0.062	0.117	0.073	_	_
	*p*	<0.01	NS	<0.05	NS	_	_
ADF	Control	56.30 ^a^	52.28 ^a^	43.67 ^Bb^	33.76 ^Bc^	2.632	<0.01
	Glucose	58.03 ^a^	54.83 ^a^	46.61 ^ABb^	41.6 ^Ab^	2.029	<0.01
	Molasses	56.62 ^a^	53.65 ^a^	49.50 ^Ab^	38.60 ^Ac^	2.073	<0.01
	Sucrose	58.13 ^a^	53.65 ^b^	50.95 ^Ab^	38.48 ^Ac^	2.210	<0.01
	Soluble starch	56.23 ^a^	50.82 ^b^	50.19 ^Ab^	33.48 ^Bc^	2.596	<0.01
	SEM	0.347	0.471	0.817	0.879	_	_
	*p*	NS	NS	<0.01	<0.01	_	_
NDF	Control	85.87 ^a^	75.99 ^Ba^	64.52 ^Cb^	53.51 ^Cd^	3.263	<0.01
	Glucose	87.52 ^a^	82.94 ^Ab^	78.58 ^Ac^	62.29 ^ABd^	2.878	<0.01
	Molasses	85.71 ^a^	80.73 ^ABb^	77.63 ^Ac^	65.58 ^Ac^	2.241	<0.01
	Sucrose	87.16 ^a^	77.02 ^Bb^	73.63 ^Bc^	60.51 ^Bd^	2.885	<0.01
	Soluble starch	87.37 ^a^	77.77 ^Bb^	74.54 ^ABc^	58.98 ^Bc^	3.086	<0.01
	SEM	0.281	0.741	1.364	1.099	_	_
	*p*	NS	<0.01	<0.01	<0.01	_	_

Mean values with different superscripts (A–D) in the same row differ within sampling days (*p* < 0.01). Means with different superscripts (a–d) in the same column differ within the treatments (*p* < 0.05). SEM, standard error of means. NS, not significant.

**Table 2 animals-13-00888-t002:** Chemical composition of rice straw fermented by white rot fungi with the addition of different carbon sources (g/kg DM) after 0, 7, 14, and 21 days.

Item	Treatment	Sampling Day				SEM	*p*
		0	7	14	21		
DM	Control	78.27 ^a^	72.71 ^Bb^	65.12 ^c^	63.29 ^ABc^	1.839	<0.01
	Glucose	79.04 ^a^	73.53 ^Bb^	61.54 ^c^	61.00 ^Bc^	2.357	<0.01
	Molasses	80.75 ^a^	71.82 ^Bb^	61.6 ^c^	61.63 ^ABc^	2.802	<0.01
	Sucrose	78.67 ^a^	75.62 ^Ab^	63.45 ^c^	64.48 ^ABc^	2.032	<0.01
	Soluble starch	80.28 ^a^	74.36 ^ABb^	63.31 ^c^	65.17 ^Ac^	2.567	<0.01
	SEM	0.314	0.380	0.455	0.501	_	_
	*p*	NS	<0.01	NS	<0.01	_	_
CP	Control	2.90 ^c^	2.99 ^Bbc^	3.15 ^BCb^	3.41 ^Ca^	0.06	<0.01
	Glucose	2.93 ^c^	3.09 ^ABbc^	3.24 ^Bb^	3.68 ^Aa^	0.09	<0.01
	Molasses	2.95 ^c^	3.29 ^Ab^	3.50 ^Aa^	3.56 ^Ba^	0.07	<0.01
	Sucrose	2.95 ^bc^	2.86 ^Bc^	3.02 ^Db^	3.32 ^Ca^	0.05	<0.01
	Soluble starch	2.97	2.95 ^B^	3.03 ^CD^	3.16 ^D^	0.04	NS
	SEM	0.02	0.05	0.05	0.05	_	_
	*p*	NS	<0.05	<0.01	<0.01	_	_
Ash	Control	9.38 ^Ac^	10.86 ^ABa^	10.99 ^Ca^	10.33 ^Cb^	0.197	<0.01
	Glucose	9.52 ^Ac^	10.63 ^Bc^	12.18 ^Aa^	11.86 ^ABa^	0.332	<0.01
	Molasses	7.47 ^Cc^	11.71 ^Ab^	11.81 ^ABab^	12.47 ^Aa^	0.600	<0.01
	Sucrose	8.43 ^Bc^	10.51 ^Bb^	11.21 ^BCab^	11.92 ^ABa^	0.402	<0.01
	Soluble starch	7.61 ^BCc^	10.76 ^Bb^	10.84 ^Cab^	11.53 ^Ba^	0.461	<0.01
	SEM	0.238	0.132	0.146	0.196	_	_
	*p*	<0.01	<0.01	<0.01	<0.01	_	_
ADF	Control	66.80 ^ABa^	58.89 ^BCb^	56.92 ^Ac^	41.24 ^Cd^	2.799	<0.01
	Glucose	65.98 ^ABa^	63.89 ^Ab^	55.97 ^Ac^	49.29 ^Ad^	2.005	<0.01
	Molasses	66.93 ^ABa^	63.66 ^Ab^	58.24 ^Ac^	50.83 ^Ad^	1.865	<0.01
	Sucrose	67.99 ^Aa^	57.64 ^Cb^	52.74 ^Bc^	44.86 ^Bd^	2.538	<0.01
	Soluble starch	64.28 ^Ba^	60.25 ^Bb^	57.43 ^Ab^	42.30 ^Bc^	2.521	<0.01
	SEM	0.399	0.697	0.538	1.042	_	_
	*p*	<0.05	<0.01	<0.01	<0.01	_	_
NDF	Control	84.61 ^ABa^	74.37 ^Cb^	71.58 ^Bc^	53.07 ^d^	3.445	<0.01
	Glucose	81.95 ^Ba^	79.17 ^Aab^	76.76 ^Ab^	63.32 ^c^	2.180	<0.01
	Molasses	86.49 ^Aa^	78.79 ^ABb^	75.75 ^Ac^	60.59 ^d^	2.849	<0.01
	Sucrose	83.93 ^ABa^	76.44 ^ABCb^	71.9 ^Bbc^	67.98 ^c^	1.831	<0.01
	Soluble starch	83.35 ^ABa^	75.90 ^BCb^	74.07 ^ABb^	57.36 ^c^	2.875	<0.01
	SEM	0.484	0.532	0.604	1.396	_	_
	*p*	<0.05	<0.01	<0.01	<0.01	_	_

Note: same as Table 1.

**Table 3 animals-13-00888-t003:** Cellulose, hemicellulose, and lignin content of corn straw.

Item	Treatment	Sampling Day				SEM	*p*
		0	7	14	21		
Cellulose	Control	29.40 ^a^	28.67 ^Aa^	18.20 ^Bb^	10.74 ^Cb^	2.367	<0.01
	Glucose	30.34 ^a^	32.07 ^ABa^	25.46 ^Aab^	22.25 ^Ab^	1.284	<0.01
	Molasses	29.21 ^a^	27.55 ^ABCa^	25.56 ^Aa^	16.44 ^Bb^	1.530	<0.01
	Sucrose	29.78 ^a^	26.37 ^BCa^	23.71 ^Ac^	16.27 ^Bd^	1.512	<0.01
	Soluble starch	28.11 ^a^	23.45 ^Ca^	25.13 ^Aa^	8.99 ^Cb^	2.268	<0.01
	SEM	0.342	0.830	0.835	1.316	_	_
	*p*	NS	<0.01	<0.01	<0.01	_	_
Hemicellulose	Control	29.23 ^a^	23.72 ^ABa^	20.85 ^Cb^	19.75 ^Bb^	1.196	<0.01
	Glucose	29.48 ^a^	28.12 ^Aab^	31.97 ^Aa^	20.69 ^ABb^	1.405	<0.01
	Molasses	29.08	27.08 ^AB^	28.13 ^AB^	26.98 ^A^	0.415	NS
	Sucrose	29.04 ^a^	23.37 ^Bb^	22.69 ^BCb^	22.03 ^ABb^	0.926	<0.01
	Soluble starch	31.14 ^a^	26.94 ^ABb^	24.35 ^Bb^	25.5 ^ABb^	0.833	<0.01
	SEM	0.334	0.616	1.149	0.878	_	_
	*p*	NS	<0.05	<0.01	<0.01	_	_
Lignin	Control	21.12 ^Ba^	19.54 ^Bb^	18.07 ^Bc^	17.54 ^ABc^	0.424	<0.01
	Glucose	18.35 ^Ca^	17.34 ^Ba^	15.61 ^Cb^	13.72 ^Cc^	0.538	<0.01
	Molasses	22.78 ^Aa^	20.58 ^Aa^	18.16 ^Bb^	16.32 ^Bb^	0.758	<0.01
	Sucrose	23.22 ^Aa^	21.77 ^Ab^	21.13 ^Ab^	16.71 ^Bc^	0.740	<0.01
	Soluble starch	23.72 ^Aa^	21.75 ^Ab^	20.00 ^ABc^	18.48 ^Ad^	0.601	<0.01
	SEM	0.528	0.508	0.518	0.443	_	_
	*p*	<0.01	<0.01	<0.01	<0.01	_	_

Note: same as Table 1.

**Table 4 animals-13-00888-t004:** Cellulose, hemicellulose, and lignin contents of rice straw.

Item	Treatment	Sampling Day				SEM	*p*
		0	7	14	21		
Cellulose	Control	41.85 ^ABa^	34.85 ^Bb^	33.31 ^Ab^	19.34 ^Bc^	2.467	<0.01
	Glucose	37.33 ^Ca^	35.86 ^ABa^	25.82 ^Cb^	24.50 ^Ab^	1.750	<0.01
	Molasses	41.82 ^ABa^	34.57 ^Bb^	29.32 ^Bc^	21.29 ^ABd^	2.279	<0.01
	Sucrose	39.32 ^BCa^	27.92 ^Cb^	24.412 ^Cc^	19.45 ^Bd^	2.215	<0.01
	Soluble starch	43.25 ^Aa^	37.35 ^Ab^	34.57 ^Ab^	19.13 ^Bc^	2.702	<0.01
	SEM	0.610	0.879	1.090	0.600	_	_
	*p*	<0.01	<0.01	<0.01	<0.01	_	_
Hemicellulose	Control	17.81 ^ABa^	15.48 ^a^	14.66 ^Bab^	11.84 ^Bb^	0.690	<0.01
	Glucose	15.96 ^Bb^	15.28 ^b^	20.79 ^Aa^	14.02 ^Bb^	0.831	<0.01
	Molasses	19.56 ^ABa^	15.13 ^ab^	17.51 ^ABab^	19.76 ^Bb^	1.203	<0.01
	Sucrose	15.93 ^Bb^	18.81 ^ab^	19.16 ^ABab^	23.12 ^Aa^	0.913	<0.01
	Soluble starch	19.07 ^ABa^	15.65 ^ab^	16.65 ^ABab^	15.06 ^Bb^	0.553	<0.01
	SEM	0.470	0.493	0.668	1.280	_	_
	*p*	<0.01	NS	<0.01	<0.01	_	_
Lignin	Control	15.57 ^Da^	13.18 ^Db^	12.61 ^Bb^	11.56 ^Cc^	0.450	<0.01
	Glucose	19.15 ^Ba^	17.41 ^Ba^	17.97 ^Aa^	12.93 ^BCb^	0.746	<0.01
	Molasses	17.64 ^Ca^	17.38 ^Bb^	17.10 ^Ac^	17.07 ^Ac^	0.072	<0.01
	Sucrose	20.25 ^Aa^	19.2 ^Ab^	17.12 ^Ac^	13.50 ^Bd^	0.778	<0.01
	Soluble starch	13.42 ^Ea^	12.14 ^Eb^	12.01 ^Bb^	11.64 ^Cc^	0.204	<0.01
	SEM	0.659	0.727	0.689	0.546	_	_
	*p*	<0.01	<0.01	<0.01	<0.01	_	_

Note: Same as Table 1.

**Table 5 animals-13-00888-t005:** In vitro fermentation characteristics of corn stover.

Item	Treatment	Sampling Day				SEM	*p*
		0	7	14	21		
IVDMD, %	Control	13.75 ^Cb^	13.53 ^Eb^	14.77 ^Ca^	15.20 ^Da^	0.2249	<0.01
	Glucose	13.99 ^Bc^	26.37 ^Bb^	32.55 ^Aa^	30.11 ^ABab^	2.1762	<0.01
	Molasses	13.95 ^BCc^	30.31 ^Ab^	33.32 ^Aa^	33.46 ^Aa^	2.4387	<0.01
	Sucrose	14.72 ^Ad^	20.52 ^Dc^	24.29 ^Bb^	24.97 ^Ca^	1.2797	<0.01
	Soluble starch	14.29 ^ABc^	22.39 ^Cb^	25.65 ^Bb^	28.03 ^BCa^	1.5540	<0.01
	SEM	0.097	1.516	1.516	1.691	_	_
	*p*	<0.01	<0.01	<0.01	<0.01	_	_
Total VFA concentration, mmol/L	Control	18.39 ^Cd^	19.48 ^Cc^	21.79 ^Ea^	20.61 ^Eb^	0.3904	<0.01
	Glucose	19.35 ^Ad^	25.72 ^Ac^	35.56 ^Aa^	31.40 ^Ab^	1.8548	<0.01
	Molasses	19.61 ^Ac^	26.13 ^Ab^	30.27 ^Ba^	20.11 ^Dc^	1.3358	<0.01
	Sucrose	18.63 ^BCd^	23.42 ^Bb^	24.35 ^Da^	21.84 ^Cc^	0.6576	<0.01
	Soluble starch	19.33 ^ABd^	22.23 ^Bc^	27.50 ^Ca^	25.88 ^Bb^	0.9640	<0.01
	SEM	0.143	0.679	1.280	1.134	_	_
	*p*	<0.01	<0.01	<0.01	<0.01	_	_
pH	Control	6.67 ^b^	6.41 ^b^	6.75 ^Ba^	6.8 ^Ba^	0.0505	<0.01
	Glucose	6.69 ^ab^	6.39 ^c^	6.65 ^Bb^	6.9 ^Ba^	0.0584	<0.01
	Molasses	6.75 ^a^	6.38 ^b^	6.74 ^Ba^	6.83 ^Bb^	0.0536	<0.01
	Sucrose	6.68 ^b^	6.49 ^c^	7.11 ^Aa^	6.66 ^Cbc^	0.0700	<0.01
	Soluble starch	6.72 ^b^	6.34 ^c^	7.06 ^Aa^	7.30 ^Abc^	0.1121	<0.01
	SEM	0.024	0.023	0.050	0.059	_	_
	*p*	NS	NS	<0.01	<0.01	_	_
NH_3_-N, mmol/L	Control	9.64 ^BCc^	13.82 ^Ea^	13.59 ^Ca^	12.5 ^Bb^	0.5023	<0.01
	Glucose	11.18 ^ABd^	16.44 ^Cb^	17.31 ^Ca^	12.28 ^Bc^	0.7920	<0.01
	Molasses	12.10 ^Ac^	18.41 ^Ab^	20.27 ^Aa^	7.75 ^Cd^	1.5085	<0.01
	Sucrose	8.76 ^Cc^	17.52 ^Ba^	13.4 ^Bb^	14.29 ^Ab^	0.9692	<0.01
	Soluble starch	9.82 ^ABd^	15.49 ^Da^	14.26 ^Db^	12.71 ^Bc^	0.6386	<0.01
	SEM	0.360	0.429	0.714	0.588	_	_
	*p*	<0.01	<0.01	<0.01	<0.01	_	_

Note: same as Table 1.

**Table 6 animals-13-00888-t006:** In vitro fermentation parameters of rice straw.

Item	Treatment	Sampling Day				SEM	*p*
		0	7	14	21		
IVDMD, %	Control	15.62 ^ABd^	26.34 ^Ea^	21.73 ^c^	23.41 ^Cb^	1.1843	<0.01
	Glucose	15.02 ^Cc^	35.55 ^Ba^	35.86 ^a^	28.49 ^Bb^	2.5478	<0.01
	Molasses	15.27 ^ABCd^	40.06 ^Aa^	29.68 ^c^	30.60 ^Ab^	4.1407	<0.01
	Sucrose	15.83 ^Ac^	28.34 ^Db^	32.72 ^a^	28.5 ^Bb^	1.9077	<0.01
	Soluble starch	15.23 ^BCd^	30.07 ^Cb^	32.69 ^a^	24.39 ^Cc^	2.0191	<0.01
	SEM	0.089	1.348	2.070	2.820	_	_
	*p*	<0.01	<0.01	NS	<0.01	_	_
Total VFA concentration, mmol/L	Control	15.46 ^Cc^	16.44 ^Db^	19.58 ^Da^	16.54 ^Cb^	0.4680	<0.01
	Glucose	16.33 ^Bd^	25.18 ^Bb^	30.68 ^Aa^	18.5 ^Bc^	1.7079	<0.01
	Molasses	17.277 ^Ac^	26.38 ^Ab^	30.33 ^ABa^	16.50 ^Cd^	1.7821	<0.01
	Sucrose	16.50 ^Bd^	22.27 ^Cb^	26.11 ^BCa^	12.4 ^Dc^	1.5883	<0.01
	Soluble starch	15.34 ^Cd^	26.1533 ^Cc^	25.16 ^Ca^	19.64 ^Ab^	1.3829	<0.01
	SEM	0.193	0.996	1.133	0.661	_	_
	*p*	<0.01	<0.01	<0.01	<0.01	_	_
pH	Control	6.68 ^b^	6.89 ^Aa^	7.04 ^Ba^	7.01 ^Aa^	0.0445	<0.01
	Glucose	6.83 ^b^	6.8 ^Ab^	7.17 ^Aa^	6.85 ^Bb^	0.0482	<0.01
	Molasses	6.77 ^a^	6.53 ^Cb^	6.64 ^Dab^	6.53 ^Db^	0.0321	<0.01
	Sucrose	6.68 ^ab^	6.53 ^Bb^	6.84 ^Ca^	6.82 ^BCa^	0.0400	<0.01
	Soluble starch	6.71 ^ab^	6.64 ^Bb^	6.71 ^Dab^	6.75 ^Ca^	0.0130	<0.01
	SEM	0.023	0.040	0.054	0.042	_	_
	*p*	NS	<0.01	<0.01	<0.01	_	_
NH_3_-N, mmol/L	Control	9.11 ^Ca^	9.49^D a^	7.55 ^Eb^	5.55 ^Ec^	0.4691	<0.01
	Glucose	9.62 ^Bb^	12.47 ^Ca^	12.42 ^Ba^	9.52 ^Ab^	0.4344	<0.01
	Molasses	10.40 ^Ab^	14.57 ^Aa^	14.34 ^Aa^	7.46 ^Cc^	0.8902	<0.01
	Sucrose	8.71 ^Dc^	12.37 ^Ca^	10.82 ^Cb^	6.47 ^Dd^	0.6710	<0.01
	Soluble starch	9.60 ^Bb^	13.61 ^Ba^	9.23 ^Dc^	8.23 ^Bd^	0.6183	<0.01
	SEM	0.154	0.457	0.635	0.369	_	_
	*p*	<0.01	<0.01	<0.01	<0.01	_	_

Note: same as Table 1.

**Table 7 animals-13-00888-t007:** Proportion of volatile acids in corn stover.

Item	Treatment	Sampling Day				SEM	*p*
		0	7	14	21		
Acetic acid, %	Control	70.00 ^Ac^	73.03 ^Cb^	75.09 ^Ca^	74.90 ^Ca^	0.628	<0.01
	Glucose	69.00 ^Bc^	75.33 ^Ab^	77.29 ^Ba^	76.84 ^Aa^	1.008	<0.01
	Molasses	68.90 ^Bc^	75.10 ^ABb^	78.40 ^Aa^	78.29 ^Aa^	1.169	<0.01
	Sucrose	70.50 ^Ac^	73.84 ^BCb^	75.00 ^Ca^	75.24 ^Ca^	0.580	<0.01
	Soluble starch	68.80 ^Bc^	74.14 ^Bb^	75.93 ^Ca^	75.72 ^Ca^	0.887	<0.01
	SEM	0.189	0.243	0.329	0.313	_	_
	*p*	<0.01	<0.01	<0.01	<0.01	_	_
Propionic acid, %	Control	18.49 ^BCc^	18.72 ^Bb^	19.18 ^Bb^	20.17 ^a^	0.210	<0.01
	Glucose	19.33 ^Ab^	19.76 ^Ab^	20.81 ^Ab^	23.25 ^a^	0.536	<0.05
	Molasses	18.24 ^Cd^	19.06 ^ABb^	19.89 ^ABb^	20.78 ^a^	0.29516	<0.01
	Sucrose	18.95 ^ABb^	18.79 ^Bb^	19.69 ^ABab^	21.52 ^a^	0.426	NS
	Soluble starch	19.27 ^Ab^	19.27 ^ABb^	19.30 ^Bb^	21.92 ^a^	0.40391	<0.05
	SEM	0.120	0.116	0.182	0.443	_	_
	*p*	<0.01	<0.01	<0.01	NS	_	_
Butyric acid, %	Control	7.25 ^ABb^	7.33 ^Bb^	8.04 ^Ca^	8.16 ^Da^	0.128	<0.01
	Glucose	6.58 ^Bc^	7.76 ^Bc^	8.81 ^Ba^	8.92 ^Ba^	0.293	<0.01
	Molasses	6.51 ^Bc^	8.69 ^Ab^	9.4.0 ^Aa^	9.51 ^Aa^	0.364	<0.01
	Sucrose	7.37 ^Aa^	6.55 ^Cb^	7.43 ^Da^	7.52 ^Da^	0.132	<0.01
	Soluble starch	6.60 ^Bb^	7.80 ^Bb^	8.21 ^Ca^	8.30 ^Ca^	0.217	<0.01
	SEM	0.116	0.191	0.185	0.186	_	_
	*p*	<0.01	<0.01	<0.01	<0.01	_	_
Valeric acid, %	Control	0.84 ^A^	0.85 ^BC^	0.84	0.85 ^B^	0.0045	NS
	Glucose	0.78 ^BC^	0.90 ^A^	1.07	0.96 ^A^	0.0427	NS
	Molasses	0.75 ^Cb^	0.85 ^Cb^	1.52 ^a^	0.93 ^Ab^	0.0958	<0.01
	Sucrose	0.80 ^B^	0.86 ^BC^	1.53	0.87 ^B^	0.1724	NS
	Soluble starch	0.85 ^A^	0.86 ^B^	0.86	0.86 ^B^	0.0042	NS
	SEM	0.011	0.005	0.146	0.012	_	_
	*p*	<0.01	<0.01	NS	<0.01	_	_

Note: same as Table 1.

**Table 8 animals-13-00888-t008:** Proportion of volatile acids in rice straw.

Item	Treatment	Sampling Day				SEM	*p*
		0	7	14	21		
Acetic acid, %	Control	65.16 ^c^	77.93 ^b^	79.99 ^Ca^	79.80 ^Ca^	1.856	<0.01
	Glucose	65.85 ^c^	80.23 ^b^	82.19 ^Ba^	81.74 ^Ba^	2.044	<0.01
	Molasses	64.04 ^c^	80.00 ^b^	83.30 ^Aa^	83.19 ^Aa^	2.403	<0.01
	Sucrose	64.70 ^b^	78.74 ^b^	79.90 ^Ca^	80.14 ^Ca^	1.959	<0.01
	Soluble starch	64.85 ^c^	79.04 ^b^	80.83 ^Ca^	80.62 ^BCa^	2.019	<0.01
	SEM	0.188	0.220	0.316	0.301	_	_
	*p*	NS	<0.01	<0.01	<0.01	_	_
Propionic acid, %	Control	19.70 ^Bc^	21.68 ^Bb^	22.22 ^Ba^	22.37 ^Ca^	0.33	<0.01
	Glucose	22.47 ^Ab^	22.66 ^Ab^	23.91 ^Aa^	24.04 ^Aa^	0.26	<0.05
	Molasses	21.92 ^Ab^	22.02 ^ABb^	22.92 ^ABa^	22.97 ^BCa^	0.16	<0.01
	Sucrose	20.29 ^Bc^	21.69 ^Bb^	22.80 ^ABa^	23.15 ^Ba^	0.35	<0.01
	Soluble starch	21.72 ^Ac^	22.16 ^ABb^	22.40 ^Bb^	22.85 ^BCa^	0.13	<0.01
	SEM	0.286	0.112	0.189	0.156	_	_
	*p*	<0.01	<0.01	<0.05	<0.01	_	_
Butyric acid, %	Control	9.06 ^ABb^	9.41 ^Dc^	10.45 ^Ca^	10.17 ^Ea^	0.176	<0.01
	Glucose	8.39 ^Bd^	10.07 ^Bc^	12.61 ^Aa^	12.17 ^Bb^	0.512	<0.01
	Molasses	8.32 ^Bc^	11.42 ^Ab^	11.52 ^Bb^	12.56 ^Aa^	0.481	<0.01
	Sucrose	9.19 ^Ab^	9.29 ^Db^	9.39 ^Db^	10.62 ^Da^	0.191	<0.01
	Soluble starch	8.41 ^Bd^	10.83 ^Ca^	11.26 ^Ba^	11.45 ^Ca^	0.375	<0.01
	SEM	0.121	0.225	0.293	0.244	_	_
	*p*	<0.05	<0.01	<0.01	<0.01	_	_
Valeric acid, %	Control	1.01 ^c^	1.10 ^bc^	1.19 ^Bb^	1.21 ^Ba^	0.0296	<0.05
	Glucose	0.94 ^b^	0.99 ^ab^	1.28 ^Aa^	1.28 ^Aa^	0.0553	<0.01
	Molasses	0.91 ^b^	1.09 ^a^	1.25 ^Ba^	1.27 ^Aa^	0.0482	<0.01
	Sucrose	0.96 ^b^	0.95 ^b^	1.19 ^Ba^	1.21 ^Ba^	0.0471	<0.05
	Soluble starch	1.01	1.11	1.20 ^C^	1.20 ^B^	0.0322	NS
	SEM	0.030	0.034	0.010	0.010	_	_
	*p*	NS	NS	<0.01	<0.01	_	_

Note: same as Table 1.

## Data Availability

The datasets generated and/or analyzed during the conduct of the study are included in this published article. Upon reasonable request, the datasets of this study are available from the corresponding author.

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
