# Peer review of "Screening the Carbon Source Type in Solid-State Fermentation with Phanerochaete chrysosporium to Improve the Forage Value of Corn Straw and Rice Straw"

_animals, 2023, doi:10.3390/ani13050888_

Round 1

Reviewer 1 Report

This paper gives the interesting results of improving nutritional quality of wheat and rice straw for feeding ruminants by solid-state fermentation with white rot fungi when four different sources of carbon are used for enhancing fermentation.

The layout of the paper is sufficiently good.  The English is also sufficiently good. The methodology is clear and detailed but requires some minor additional explanations. The results are presented in manner so that potential readers can clearly follow the given discussion. However , my main concern is that authors claimed that the aim was to find an optimal fermentation conditions and even word Optimizing is present in the title, but there is no any optimization process presented in this work nor any optimal result is given. There is complete lack of any model or statistics that support the process of optimization there are only presented results and descriptive statistics together with ANOVA. The authors need to add optimization process or to completely exclude claims that this experiment was optimization. There is claims that carbon source was optimized, but on the results only show which carbon source acts as a better fermentation promoter. My other comments are as follows:

Lines 18-20: The statement from this sentence is already presented in the first sentence of summary. This should be deleted or largely rephrased.

Line 20: Delete the word groups.

Lines 41-42: Please provide the reference for claim regarding statistics.

Lines 45-46: Some fibers such as? Please be more specific, and provide a suitable reference if necessary.

Lines 84-86: Is this sentence meant to be put into conclusion section? If authors put this claim here, where is the reference to previous work, or why do even bother do the investigation in first place if carbon source is known to enhance fermentation?

Lines 96-98: Please divide this sentence into two.

Lines 105-106: Please give better explanation of this part. Does this mean that you approximated moisture content or you measure it somehow for at least one or more samples?

Line 108: Remove this sentence, you say this in the next section of the manuscript.

Line 118: Which fermentation parameters were measured? Give the list here, not later in the text.

Line 200: There is no figure 1.2 in this manuscript. Perhaps you mean 1A?

Line 226: The numbers on y-axis increase over time thus I reckon that authors presented loss rate and not losses. Thus, please make corrections in figure title. Also, please add letters A, B and C in brackets next to the cellulose, hemicellulose and lignin, respectively.

Line 229: Same as previous.

Author Response

Response to Reviewer 1 Comments

Dear Reviewer:

We are very pleased to have been given the opportunity to revise our manuscript entitled for “Optimizing the Carbon Source Type in Solid-State Fermentation with Phanerochaete chrysosporium to Improve the Forage Value of Corn Straw and Rice Straw” (Manuscript Number: animals-2202243). We want to extend our appreciation to you and the reviewers for taking the time and effort necessary to provide such insightful guidance. We have carefully considered the comments offered by the reviewers. Herein, we explain how we revised the paper based on those comments and recommendations. The manuscript has certainly benefited from these revision suggestions. Corresponding changes have been made in our manuscript and are marked in yellow. The main corrections in the paper and the responses to the reviewer’s comments are as flowing.

Point 1: Lines 18-20: The statement from this sentence is already presented in the first sentence of

summary. This should be deleted or largely rephrased.

Response 1: Thank you for your valuable comments, we deleted the sentence as you suggested (Lines 18-20). Thank you very much again.

Point 2: Line 20: Delete the word groups.

Response 2: Thank you for the comment, that part has been deleted (Line 18). Thank you very much again.

Point 3: Lines 41-42: Please provide the reference for claim regarding statistics.

Response 3: Thank you for your comment. We have added a reference to the data source (Lines 42). Thank you very much again.

Point 4: Lines 45-46: Some fibers such as? Please be more specific, and provide a suitable reference if necessary.

Response 4: Thank you for the comment. We tried to convey a clear meaning by correcting the part where the meaning of the sentence was vaguely reflected, and the sentence was modified as follows: (Lines 46-47). Thank you very much again.

Point5: Lines 84-86: Is this sentence meant to be put into conclusion section? If authors put this claim here, where is the reference to previous work, or why do even bother do the investigation in first place if carbon source is known to enhance fermentation?

Response 5: Thank you for your comment. This sentence should have appeared as a conclusion and we have amended it (Lines 85-86). Thank you very much again.

Point 6: Lines 96-98: Please divide this sentence into two.

Response 6: Thank you for your comment. We changed the sentence as you suggested (Lines 97-99). Thank you very much again.

Point 7: Lines 105-106: Please give better explanation of this part. Does this mean that you approximated moisture content or you measure it somehow for at least one or more samples?

Response 7: Thank you for pointing out the details. We have amended and added clarification to the section accordingly (Lines 100-103). Thank you very much again.

Point 8: Line 108: Remove this sentence, you say this in the next section of the manuscript.

Response 8: Thank you for your comment, we have removed this sentence and stated it in the next section of the manuscript (Lines 113-115). Thank you very much again.

Point 9: Line 118: Which fermentation parameters were measured? Give the list here, not later in the text.

Response 9: Thank you for your comment, we have added a list of measured fermentation parameters (Line119-120). Thank you very much again.

Point 10: Line 200: There is no figure 1.2 in this manuscript. Perhaps you mean 1A?

Response 10: Thank you for your comment, we overlooked that detail. We have made a correction (Line 203-204). Thank you very much again.

Point 11: Line 226: The numbers on y-axis increase over time thus I reckon that authors presented loss rate and not losses. Thus, please make corrections in figure title. Also, please add letters A, B and C in brackets next to the cellulose, hemicellulose and lignin, respectively.

Response 11: Thank you for the comment. We accepted the comments and modified them to clearly convey the meaning (Lines 228-231). Thank you very much again.

Point 12: Line 229: Same as previous.

Response 12: Thank you for the comment. We accepted the comments and modified them to clearly convey the meaning (Lines 221-234). Thank you very much again.

Reviewer 2 Report

The results in this paper are clear. Sugars are known to repress polymers degradation.

However the rational of adding expensive carbon source (sugars) to agricultural waste materials is not clear and the digestibility remains low. Please explain.

The authors must change "white rot bacteria, P. chrysosporium"  to fungus, this mistake appears several times (many better references  then 20). In addition, P. chrysosporium is indeed a white rot, but one can not generalize the results to other white rot fungi as the results can be different for each fungus.

Author Response

Response to Reviewer 2 Comments

Dear Reviewer:

Thank you for your letter and for the reviewers’ comments concerning our manuscript entitled “Optimizing the Carbon Source Type in Solid-State Fermentation with Phanerochaete chrysosporium to Improve the Forage Value of Corn Straw and Rice Straw”(Manuscript Number: animals-2202243). Those comments are all valuable and very helpful for revising and improving our paper, as well as the important guiding significance to our research. We have studied the comments carefully and have made a correction which we hope meets with approval Revised portion is marked in green in the paper. The main corrections in the paper and the responses to the reviewer’s comments are as flowing.

Point 1:The results in this paper are clear. Sugars are known to repress polymers degradation.

However the rational of adding expensive carbon source (sugars) to agricultural waste materials is not clear and the digestibility remains low. Please explain.

Response 1: Thank you for your constructive comments. Because carbon sources are the nutrients that make up the source of carbon in microbial cells and metabolites. The addition of a carbon source can help the white rot fungus to convert the lignocellulose into a higher protein content feed. Although the addition of carbon did not significantly increase digestibility, it did increase the protein content and lignin degradation rate. Thank you very much again.

Point 2:The authors must change "white rot bacteria, P. chrysosporium" to fungus, this mistake appears several times (many better references  then 20). In addition, P. chrysosporium is indeed a white rot, but one can not generalize the results to other white rot fungi as the results can be different for each fungus.

Response 2: Thank you for your constructive comments. We have overlooked this problem and have corrected the manuscript as you suggested and replaced it with a new reference (Line 88). Thank you for your advice, indeed this experiment is only representative of the effect of P. chrysosporium and not of all fungi. Thank you very much again.

Reviewer 3 Report

The objective of the manuscript titled: “Optimizing the Carbon Source Type in Solid-State Fermentation with Phanerochaete chrysosporium to Improve the Forage Value of Corn Straw and Rice Straw”

This article's objective was to evaluate to the optimal fermentation conditions for corn and rice straw with white rot fungi Phanerochaete chrysosporium to improve nutrient availability and digestibility for use in ruminant feeds. The results may be helpful for the improvement of the high quality feed production and feed digestibility by ruminant animals.

The findings of this study can be helpful for further research, which makes the results significant for the scientific peers, as well as for the food industry worldwide.

In the Introduction part of the manuscript, the subject was well addressed and citation of the literature is adequate and up to date. However, there are numerous spelling errors and language issues that need to be addressed.

The Material and Methods need to be supplemented by the correct names and manufacturers of the chemicals and instruments used in research, as well as method numbers and references. Otherwise, the methods used are appropriate for this kind of research.

The Result and the Discussion part of the article tackle the subject of the research appropriately, and elaborate the findings well, by making appropriate conclusions and references to other previously conducted researches.

The References are up to date and refer to the subject correctly.

My comments are:

1.      I suggest the English language be checked for corrections.

2.      Correct spelling and word spacing.

3.      Supplement the Materials and Methods as suggested.

4.      Use italics for Latin terms.

5.      Use synonyms when a word is repeated within the sentence multiple times, if possible.

The article needs revision.

My comments are included in the pdf Manuscript file.

Author Response

Response to Reviewer 3 Comments

Dear Reviewer:

Thank you for your letter and for the reviewers’ comments concerning our manuscript entitled “Optimizing the Carbon Source Type in Solid-State Fermentation with Phanerochaete chrysosporium to Improve the Forage Value of Corn Straw and Rice Straw” (Manuscript Number: animals-2202243). Those comments are all valuable and very helpful for revising and improving our paper, as well as the important guiding significance to our research. We have studied the comments carefully and have made a correction which we hope meets with approval Revised portion is marked in red in the paper. The main corrections in the paper and the responses to the reviewer’s comments are as flowing.

Point 1: I suggest the English language be checked for corrections.

Response 1: Thank you for your careful reading of our manuscript. We changed the sentence as you suggested. Thank you very much again.

Point 2: Correct spelling and word spacing.

Response 2: Thank you for your careful reading of our manuscript. We have overlooked these important details and we have corrected the details of word spelling and word distance. Thank you very much again.

Point 3: Supplement the Materials and Methods as suggested.

Response 3: Thank you for your valuable comments. We changed the sentence as you suggested. Thank you very much again.

Point 4: Use italics for Latin terms.

Response 4: Thank you for the comment, we changed the sentence as you suggested. Thank you very much again.

Point 5: Use synonyms when a word is repeated within the sentence multiple times, if possible.

Response 5: Thank you for your careful reading of our manuscript. We changed the sentence as you suggested. Thank you very much again.

Round 2

Reviewer 1 Report

The authors make necessary changes and I would like to recommend the manuscript for publishing.

Author Response

Response to Reviewer 1 Comments

Dear Reviewer:

Thank you very much for your valuable review comments on the articles I have submitted. I take your review comments very seriously and have made changes based on your suggestions to make the article more complete and rigorous. I have benefited greatly from your valuable comments in your article. Thank you for approving my manuscript for publication. Thank you very much again.

Reviewer 2 Report

the cost/ benefit (sugar vs protein)   should be discussed  and calculated 

Author Response

Response to Reviewer 2 Comments

Dear Reviewer:

We are very pleased to have been given the opportunity to revise our manuscript entitled for “Screening the Carbon Source Type in Solid-State Fermentation with Phanerochaete chrysosporium to Improve the Forage Value of Corn Straw and Rice Straw” (Manuscript Number: animals-2202243). Thank you for your decision and constructive comments on my manuscript. We want to extend our appreciation to you and the reviewers for taking the time and effort necessary to provide such insightful guidance. We have carefully considered the comments made by the reviewers and have provided the following responses.

Point 1: the cost/ benefit (sugar vs protein) should be discussed and calculated Suggestions for Authors.

Response 1: Thank you, for your constructive comments. This issue is indeed one that we have overlooked. Although the addition of sugar shortens the fermentation time and improves the palatability and crude protein content of the straw feed, it also increases the cost of the feed. We feel that we should follow up with animal feeding trials to assess the economic benefits of this method. The fermentation conditions are optimised to reduce the amount of sugar added or to find alternatives to sugar in the form of food waste (e.g. apple pomace, pear pomace, etc.). Thank you very much again.